# Socioeconomic inequalities in utilizing maternal health care in five South Asian countries: A decomposition analysis

Md. Ashfikur Rahman [1]*, Md. Amirul Islam[1], Mortuja Mahamud Tohan[1], S. M. Muhibullah[2], Md. Sazedur Rahman[2], Md. Hasan Howlader[1]

1 Development Studies Discipline, Social Science School, Khulna University, Khulna, Bangladesh,
2 Statistics Discipline, Science, Engineering and Technology School, Khulna University, Khulna, Bangladesh

* ashfikur@ds.ku.ac.bd

## Abstract

**Data Availability Statement:** This study utilized the publicly available Demographic and Health Surveys (DHS) Program dataset of Bangladesh, which can

### Background

High maternal mortality rates still today remain a significant public health concern in South Asian countries. The majority of maternal deaths occur during pregnancy, and these deaths may typically be avoided by ensuring that women have access to reliable maternity care such as antenatal care (ANC) and facility delivery. The objectives of this research were to assess socioeconomic disparities in the utilization of health care services by mothers and to determine the factors influencing this utilization among women aged 15 to 49 in five South Asian countries.

### Methods

For this study, nationally representative data from the Demographic and Health Survey (DHS) were analyzed. This research included a total of 262,531 women between the ages of 15 and 49. To determine the likely causes of maternal health care utilization, simple bivariate statistics and binary logistic regression were applied, and decomposition analysis and the concentration curve were used to quantify disparity (Lorenz curve).

### Results

ANC and institutional delivery were both prevalent in 59.27% and 86.52% of cases, respectively. Among the five nations, Maldives has the greatest ANC (96.83%) and institutional delivery (99.39%), while Bangladesh has the lowest ANC (47.01%) and institutional delivery (49.81%). Women's and husbands' education, household wealth status, BMI, and urban residents are the most important factors influencing maternal health service utilization, whereas higher education level, affluent wealth quintiles, and place of residence are the major contributors to socioeconomic inequalities in access to maternal health care that favor the wealthy.

be obtained freely from https://dhsprogram.com/. As the third-party user, we do not have permission to share the data publicly on any platform. Data are accessible free of charge upon a registration with the Demographic and Health Survey program (The DHS Program).

**Funding:** This article was partially funded by Khulna University Research and Innovation Centre (RIC). The grant number is KU/RIC-04/2000-288 however, the funders had no role in study design, data collection and analysis, decision to publish, or preparation of the manuscript.

**Competing interests:** The authors have declared that no competing interests exist.

## Conclusion

Maternal health care services must be utilized properly in order to promote optimal health and prevent maternal mortality. Several socioeconomic and sociodemographic variables of the individual population, as well as policy issues, all have an impact on maternal mortality. This research recommends for concerted action to enhance how successfully women use maternity care services.

## Background

Globally, 810 maternal deaths are estimated to occur daily, most of which can be avoided between 2000 and 2017, there was a worldwide 38% decline in the MMR [1]. Complications during childbirth and pregnancy claimed the lives of 295,000 women worldwide in 2017; Lower—middle countries contributed mostly to those deaths. In 2017, 86% (254,000) of all maternal fatalities occurred in Sub-Saharan Africa and South Asia, whereas South Asia was responsible for 20% (58,000) of all maternal deaths [2]. The leading causes of high maternal mortality include the lack of antenatal care (ANC), facility deliveries, and postnatal care (PNC), as well as the absence of skilled birth attendants (SBAs) during deliveries [3–5]. World leaders set the goal to reduce MMR down to 70 per 100000 live births within 2030 through achieving sustainable development goals (SDG) [6].

According to reports, five south Asian countries, Bangladesh, Indonesia, Myanmar, Nepal, and Sri Lanka, were doing great in providing better maternal care services and average health of mothers was observed better in those countries [2]. Whereas other countries from the same region still pose a high maternal mortality rate, India decreased its MMR from 556 per 100000 live births in 1990 to 145 in 2017 [7], Pakistan has MMR of 186/100000 in 2017 live births [8] and Afghanistan has the highest MMR rate 638/100,000 live births [7]. Bangladesh, Indonesia, Myanmar, Nepal, and Sri Lanka made remarkable reductions in maternal mortality to meet the Millennium Development Goals (MDGs) [3], whereas other countries from the same region are still struggling with their own targets for Maternal Mortality rates. This achievement is primarily due to significant improvements in institutional deliveries, lower costs associated with delivery in a public health center, and initiatives to address societal disparities like increased female educational attainment, decreased fertility, and decreased household poverty [9].

Previous research has shown that inadequate maternal medical care are a major contributor to high rates of morbidity and mortality among mothers [10–12]. Women who receive antenatal care (ANC) on a regular basis throughout their pregnancy, give birth in a healthcare center, and receive postnatal care (PNC) within a week of giving birth have a lower risk of maternal death [4,12]. Utilization of services dedicated for maternal health care stands as a significant issue that has received a great deal of attention in the public health literature, and it is fundamental to examine the most influencing factor affecting health-seeking behavior and health-care service utilization, from individual to community-level factors, as well as to pay close attention to these factors [13,14].

Recent studies in India found that maternal health care service utilization is strongly correlated with factors like education, household wealth status, media exposure, religious affiliation, and women's education [10,11]. A study in Ethiopia [15] and Turkey [16] has shown that in addition to socioeconomic and cultural factors, perceived advantages and needs, as well as accessibility-related factors have an impact on how often women use maternal care services. In

a previous study of Bangladesh [14], income, the distance to the nearest medical facility, health cards or health insurance, land ownership, the sorts of latrines used, and membership in a community group have all been documented as factors that affect whether or not a woman uses maternal care services.

The World Health Organization (WHO), stated that health services usage entails providing at least basic health care to the world's poor and disadvantaged without discrimination as well as ensuring that those facilities are readily available and socially acceptable to all [17]. Reasonable efforts are urgently required to ensure sufficient access to medical care and to eliminate inequalities. This article seeks to pinpoint the determinants for utilizing maternal health services in South Asian women aged between 15–49 years using the most recent databases. The two main variables of maternal health service use—(i) At least four ANC visits and (ii) institutional delivery—were studied in this study together with their socio-demographic and economic determinants. This study will provide relevant information regarding the characteristics that influence the usage of maternal care in five South Asian nations, which may help socially disadvantaged groups better utilize these services.

## Methods

### Data source and sampling

The present study used the most recent wave of secondary datasets, including Bangladesh, India, Maldives, Nepal, and Pakistan's Demographic and Health Survey (DHS) data (BDHS 2017–18; IDHS 2019–21; MDHS 2016–17; NDHS 2018; PDHS 2017–18). The survey was carried out in accordance with national regulations. These DHS Surveys employed a multistage stratified sampling technique. Each country is divided into geographical areas, such as division, in the initial phase [18]. Within these subnational regions, different strata of the population reside in urban or rural areas. These primary sample units, or clusters, are selected with a probability proportional to each cluster's proportional contribution to the entire population. During the second sample stage, every family in the cluster is enumerated, and an average of 25 of them are selected at random for interviews using a systematic sampling procedure with equal probabilities [19]. The DHS report for the specific nation contains more information regarding the sampling procedure. The ever-married women who had children were questioned about the nutritional condition of their children.

### Outcome variables

In order to assess the sociodemographic and economic determinants that impact maternal healthcare consumption, two outcome variables were included: appropriate ANC visits during pregnancy (4 ANC visits) and institutional delivery *("whether a woman gave birth in a government hospital, district hospital, mother and child welfare center (MCWC), Upazila health complex, health, and family welfare center, private hospital/clinic, private medical college/university, rural health center, or private medical college/university"*) [4,5,8,20].

### Independent variables

The researcher identified the independent variables for this study after analyzing the most current relevant literature. The selected sociodemographic and economic independent variables included in the analysis are the ("*place of residence (urban and rural), women's age (15–24, 25–34, and 35–49 years), women's highest education level (no education, primary, secondary and higher), body mass index (BMI) (underweight: <18.50kg/m2, normal: 18.50–24.99kg/m2, overweight/obese: ≥25.00kg/m2), Women's current occupational status (Working; Not working),*

*Husband's education (no education, primary, secondary, higher); Husband's occupation (agricultural; non-agricultural) and wealth status (poorest, poorer, middle, richer, richest")* [5,9,11,13,15,21–26].

## Statistical analysis

The raw data was filtered before the analysis began, starting with data from the women between the ages of 15 and 49 years. The second stage of filtration involves removing responses that were either missing or unreported for the variables used to calculate the outcome variables. A total of 262,531 data (unweighted) were retrieved. The estimates were nationally and comparably representative because we used DHS sample weight, clustering, and stratification data [18]. Due to the reciprocal reliance of error terms within clusters and households, it was required to cluster the sample at the main sampling unit level. To account for unobservable country-level variables in pooled analyses, data were reweighted by population size and country-fixed effects were added. For the combined analysis, two distinct binary logistic regression models were developed. First, we ran separate models (single-adjusted models) for each variable of interest; next, we ran a combined model (fully adjusted model) that accounted for all relevant variables simultaneously.

The distribution of ANC seeking and institutional delivery across explanatory factors is tabulated using basic descriptive statistics. The usage of a Lorenz curve was devised to determine whether or not there is a socioeconomic inequality in women' access to antenatal care. Visualize inequity in maternal health care use by comparing the cumulative proportion of respondents who reported utilizing maternal health care to the cumulative percentage of respondents who reported having a minimal level of wealth status. Whenever there is a deviation from the line, which is marked by a 45° line, inequality is shown. When the concentration curve sits below the line of equality, people with a greater wealth level are more likely to report more health care usage, whereas those with a lower wealth status are less likely to report any health care utilization. To measure the level of concentration, the CIX was calculated. The formula was developed by Kakwani [27], Jenkins [28] and Kakwani et al., [29] was used to calculate the CIX is known as the convenient covariance approach. The formula is described as following-

$$CIX = \frac{2}{\mu}cov(h, r);$$

Description has been taken from author's previously published paper [5,30] *("where r is the fractional rank of persons in the distribution of the variable by which concentration will be computed (wealth status), μ is the weighted mean of the indicator whose concentration is to be calculated in this case (health care utilization), and h is the variable whose concentration has to be estimated; The covariance between h and r is illustrated by the equation cov(h, r). CIX recognizes numbers between -1 and 1. The closer CIX is to +1, the bigger the upper quantile concentration of the variable used to compute concentration, and the closer CIX is to -1, the greater the lower quantile concentration").*

To examine the impact of explanatory factors on the CIX, the CIX was decomposed based on all these explanatory variables. Here we used the decomposition technique proposed by O'Donnell et al. [23]. The procedure begins by fitting the subsequent regression line using the user-written STATA commands Lorenz [31] and conindex [23] to generate the Lorenz curve and measure CIX, respectively.

### Ethical approval

No additional ethical approval was needed for this study because it used publicly available secondary data from Demographic and Health Surveys (DHS) Program. The respective country's report contains specifics regarding the DHS Program's adherence to ethical standards. All the procedures were carried out in conformity with the pertinent rules and laws.

## Results

### Background characteristics

Using descriptive statistics, Table 1 illustrates the typical features of women utilizing maternal health care in five South Asian nations. The findings of 2,62,531 observations obtained in Bangladesh, India, Maldives, Nepal, and Pakistan are presented in the table. The majority of women (88.72%) were from India, and the majority were from rural regions (77.52%) and from the poorest (26.84%) and poorer (23.26%) socioeconomic groups. The 25-34-year-old age group consists of the most women (58.52%), and the majority of them had a normal BMI (62.72%). It was also observed that the majority of women had a secondary education (49.60%), however the fewest women were employed (23.94%). In addition, the most common husband's level of education was secondary (47,54%), and the majority of them were non-agricultural professionals (81,75%).

Fig 1, displays a graphical depiction of the overall and country-specific prevalence of weighted maternal health care usage. The overall prevalence of ANC and institutional births were 59.27% and 86.52%, respectively. Maldives has the greatest ANC (96.83%) and institutional delivery (99.39%) out of the five nations, whilst Bangladesh has the lowest ANC (47.01%) and institutional delivery (49.81%).

### Contributing factors to maternal health care services (Pooled data)

In the Table 2 those who lived in urban areas were 1.22 times (CI: 1.15–1.29) more likely to get ANC and 1.26 times (CI: 1.17–1.35) more likely to give birth in a hospital than those who lived in rural regions. ANC was 1.09 times (CI: 1.01–1.17) more frequent in women aged 25–34 compared to those aged 35–49 years. Women between the ages of 15 and 24 and 25 to 34 were 1.25 (CI: 1.15 to 1.36) and 1.56 (CI: 1.42–1.70) times more likely to give birth in a hospital than women between the ages of 35 and 49. Obese and overweight women were shown to have 1.18 (CI: 1.11–1.26) and 1.25 (CI: 1.12–1.39) times greater chances of receiving ANC and 1.20 (CI: 1.11–1.30) and 1.58 (CI: 1.11–1.30) times greater odds of institutional delivery, respectively (CI: 1.37–1.82). The utilization of maternal health care services was affected by both the husband and wife's levels of education. As predicted, there was a high correlation between the institutional delivery, ANC and the levels of wealth. Compared to unemployed women, working women were 1.17 times (CI: 1.10–1.23) more likely to get ANC and 0.90 times (CI: 0.85–0.96) less likely to give birth in a hospital. ANC was identified in 1.12 times (CI: 1.06–1.19) more women with spouses who worked outside the home (CI: 1.06–1.19).

### Predictors of maternal health care use in the countries investigated

In Bangladesh, urban women more commonly utilized two maternal health care services than rural women (S1 Table). In comparison to women with normal BMI, there was a considerable association between the women's BMI and their usage of ANC services and institutional delivery. Education influences the utilization of maternal health care services in a beneficial manner. Compared to those with a lower level of education, those with a higher level of education were 3.54 and 3.94 times more likely to give birth in a hospital facility. Interestingly, women

**Table 1. Background characteristics of study participants.**

| Characteristics | Number | % |
|---|---|---|
| **Country** | | |
| Bangladesh | 8,759 | 3.34 |
| India | 232,920 | 88.72 |
| Maldives | 3,106 | 1.18 |
| Nepal | 5,038 | 1.92 |
| Pakistan | 12,708 | 4.84 |
| **Place of Residence** | | |
| Urban | 59,015 | 22.48 |
| Rural | 203,516 | 77.52 |
| **Maternal Age** | | |
| 15–24 | 81,546 | 31.06 |
| 25–34 | 153,642 | 58.52 |
| 35–49 | 27,343 | 10.42 |
| **Body Mass Index** | | |
| <18.50 (Underweight) | 44,688 | 18.21 |
| 18.50–24.90 (Normal) | 153,949 | 62.72 |
| 25.00–29.99 (Overweight) | 36,172 | 14.74 |
| <30 (Obesity) | 10,626 | 4.33 |
| **Women's Highest Education Level** | | |
| No education | 60,107 | 22.90 |
| Primary | 35,941 | 13.69 |
| Secondary | 130,223 | 49.60 |
| Higher | 36,260 | 13.81 |
| **Current Working Status** | | |
| Not working | 49,616 | 76.06 |
| Working | 15,615 | 23.94 |
| **Husband's Education Level** | | |
| No education | 11,399 | 17.61 |
| Primary | 11,310 | 17.48 |
| Secondary | 30,762 | 47.54 |
| Higher | 11,242 | 17.37 |
| **Husband's Occupation** | | |
| Agricultural | 11,604 | 18.25 |
| Non-Agricultural | 51,971 | 81.75 |
| **Wealth Status** | | |
| Poorest | 70,469 | 26.84 |
| Poorer | 61,072 | 23.26 |
| Middle | 51,097 | 19.46 |
| Richer | 44,321 | 16.88 |
| Richest | 35,572 | 13.55 |

who worked in any capacity were more likely to utilize ANC services, but less likely to deliver in a hospital. Women were significantly more likely to utilize two maternal health care services if their husbands or partners had higher education than those with no education. There was a high association between household wealth and the two maternal health care services, it was discovered 3.86 (3.02–4.95) times more often than women from low-income households, women from the wealthiest households gave birth in a hospital (AOR: 2.38; 95% confidence

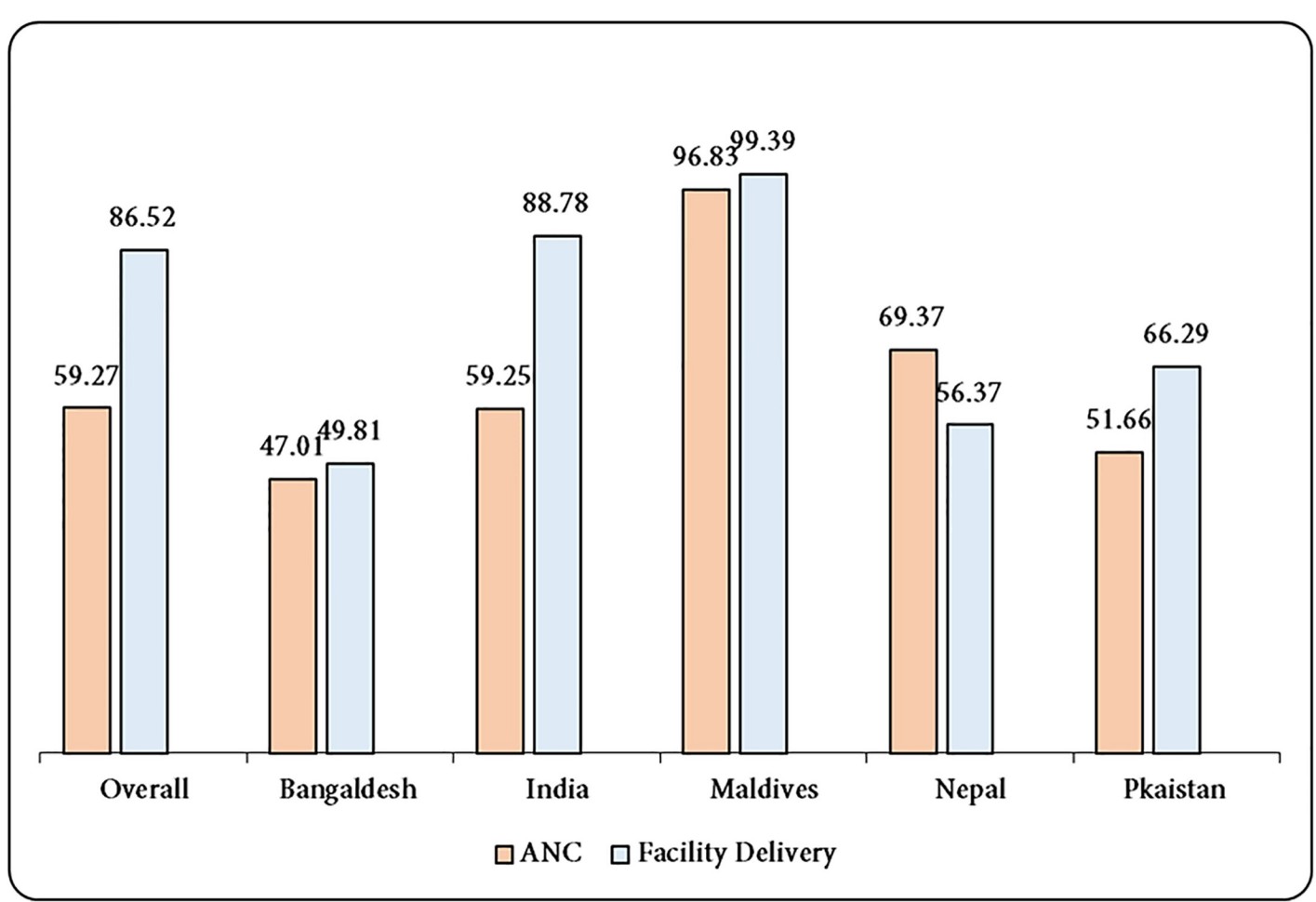

**Fig 1. Prevalence of maternal health care utilization by women (ANC and Institutional Delivery) (weighted).**

interval: 2.38 to 4.95). In contrast, low-income women were less likely to utilize the two maternity care programs.

According to study, urban women in India are 1.13 times (CI: 1.06–1.22) more likely to have ANC exams than women living in rural areas (S2 Table). Younger women were more likely than those between the ages of 35 and 49 to give birth in a hospital. Obese or overweight women were more likely to use ANC and deliver in a hospital than women with a normal BMI. Compared to less educated women and their partners, more educated women and their partners utilize ANC checks and institutional delivery more frequently. As predicted, there was a strong link between wealth and maternal health care service utilization.

In Nepal (S3 Table), women living in urban areas were 1.88 times (CI: 1.55–2.27) more likely to give birth in a hospital than women living in rural areas. The utilization of maternal health care services was positively connected with the education level of the mothers. It was found that women from more affluent families utilized ANC checks and institutional delivery.

Urban women in Pakistan (S4 Table) had 1.47 (CI: 1.22–1.78) and 1.19 (CI: 1.02–1.40) times better chances of receiving ANC schooling and having their infants delivered in a hospital than rural women. Younger women utilized two maternity care services more frequently than older mothers. Obese and overweight women were 1.24 (CI: 1.01–1.52) and 1.44 (CI:

**Table 2. Factors associated with ANC: Institutional delivery: Pooled analysis.**

| Characteristics | AOR ANC (95% CI) | AOR Institutional Delivery (95% CI) |
|---|---|---|
| **Place of Residence** | | |
| Urban | 1.22 (1.15–1.29)*** | 1.26 (1.17–1.35)*** |
| Rural (RC) | | |
| **Maternal Age** | | |
| 15–24 | 1.07 (0.98–1.15) | 1.56 (1.42–1.70)*** |
| 25–34 | 1.09 (1.01–1.17)* | 1.25 (1.15–1.36)*** |
| 35–49 (RC) | | |
| **Body Mass Index** | | |
| <18.50 (Underweight) | 0.98 (0.93–1.04) | 1.05 (0.98–1.12) |
| 18.50–24.90 (Normal) (RC) | | |
| 25.00–29.99 (Overweight) | 1.18 (1.11–1.26)*** | 1.20 (1.11–1.30)*** |
| <30 (Obesity) | 1.25 (1.12–1.39)*** | 1.58 (1.37–1.82)*** |
| **Women's Highest Education** | | |
| No education (RC) | | |
| Primary | 1.31 (1.21–1.41) *** | 1.19 (1.11–1.29) *** |
| Secondary | 1.68 (1.57–1.80) *** | 1.88 (1.76–2.02) *** |
| Higher | 2.02 (1.84–2.22) *** | 3.66 (3.20–4.18) *** |
| **Current Working Status** | | |
| Not working (RC) | | |
| Working | 1.17 (1.10–1.23) *** | 0.90 (0.85–0.96) ** |
| **Husband's Education Level** | | |
| No education (RC) | | |
| Primary | 1.15 (1.07–1.25) *** | 1.27 (1.17–1.37) *** |
| Secondary | 1.31 (1.22–1.41) *** | 1.43 (1.33–1.53) *** |
| Higher | 1.33 (1.21–1.46) *** | 1.72 (1.54–1.93) *** |
| **Husband's Occupation** | | |
| Agricultural (RC) | | |
| Non-Agricultural | 1.12 (1.06–1.19) *** | 1.00 (0.93–1.07) |
| **Wealth Status** | | |
| Poorest (RC) | | |
| Poorer | 1.25 (1.18–1.34) *** | 1.61 (1.51–1.72) *** |
| Middle | 1.70 (1.59–1.82) *** | 2.48 (2.29–2.68) *** |
| Richer | 1.93 (1.78–2.08) *** | 3.47 (3.16–3.82) *** |
| Richest | 2.54 (2.31–2.79) *** | 6.21 (5.43–7.11) *** |

*p<0.05,

**p<0.01,

***p<0.001 while AOR stands for Adjusted Odds Ratio.

1.16–1.79) times more likely to have ANC and give birth in a hospital facility than normal-BMI women. Both the woman and husband's level of education and the household's wealth were strongly related to the usage of maternal health care services.

Using a Lorenz curve technique (concentration curve; CC), Fig 2 illustrates the inequalities in ANC-seeking behavior amongst the studied countries. Four CCs go below the equality line, indicating that women from wealthy households are more likely to have ANC. For Pakistan, the distance between the line of equality and the CC was determined to be the greatest means

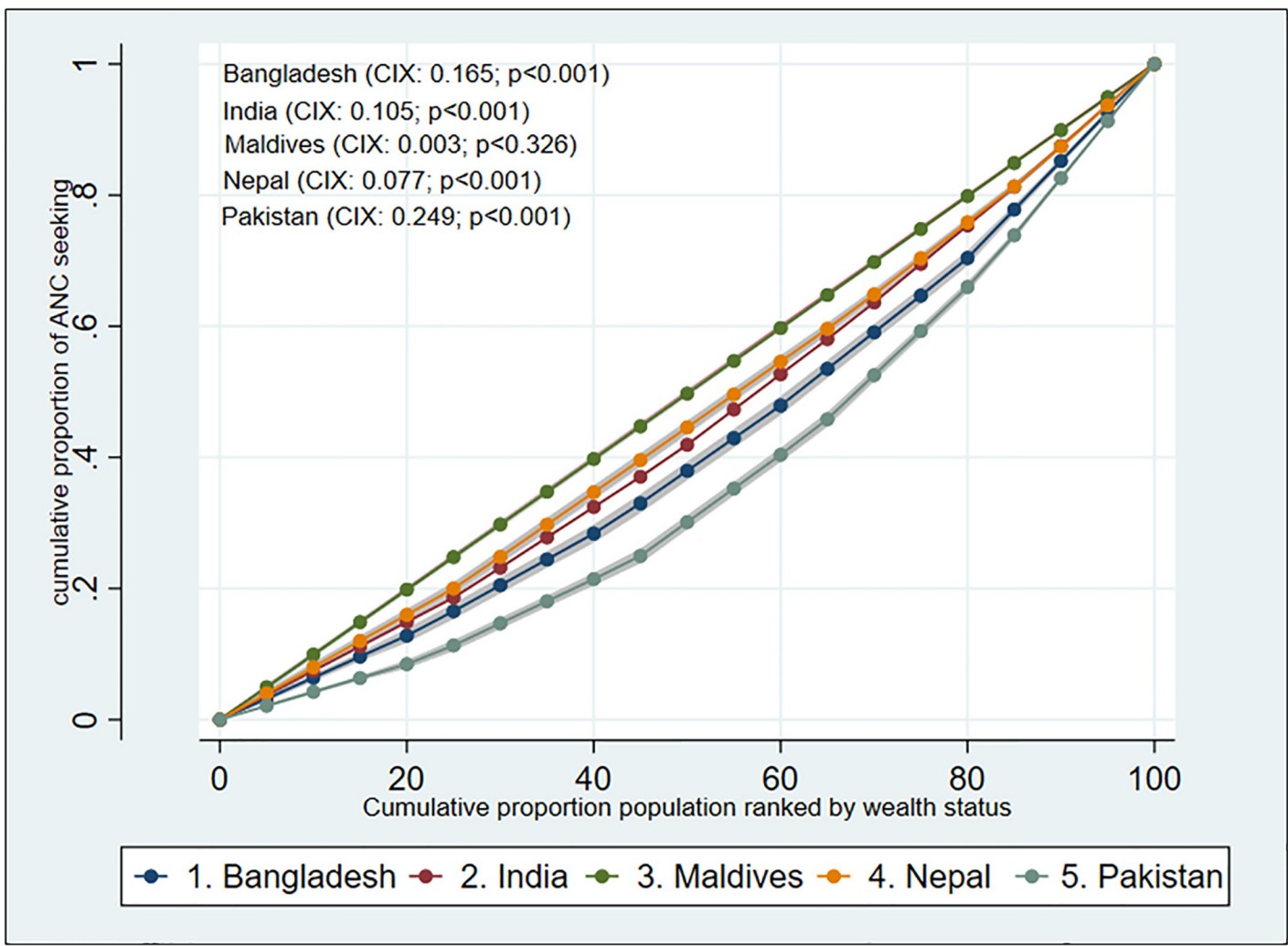

**Fig 2. Concentration curve for inequality estimation of ANC seeking behaviour.**

the extent of inequality exist much among affluent groups, whereas for the Maldives, it was the lowest.

Using a Lorenz curve, Fig 3 illustrates the inequalities in institutional delivery among the investigated nations (concentration curve; CC). The fact that four CCs are below the equality line indicates that women from rich households are more likely to birth at a hospital. It was determined that the Maldives had the smallest distance between the line of equality and the CC, whereas Pakistan had the most means inequality is less in Maldives and much among Pakistani well-off groups.

## Decomposition analysis

The impact of major socioeconomic and demographic factors on maternal health care utilization and inequalities are shown in Tables 3 and 4. The degree of change in the dependent variable, which is a socioeconomic imbalance in maternal health care utilization, that resulted from a one-unit change in the exploratory variables is shown in the "Elasticity" column.

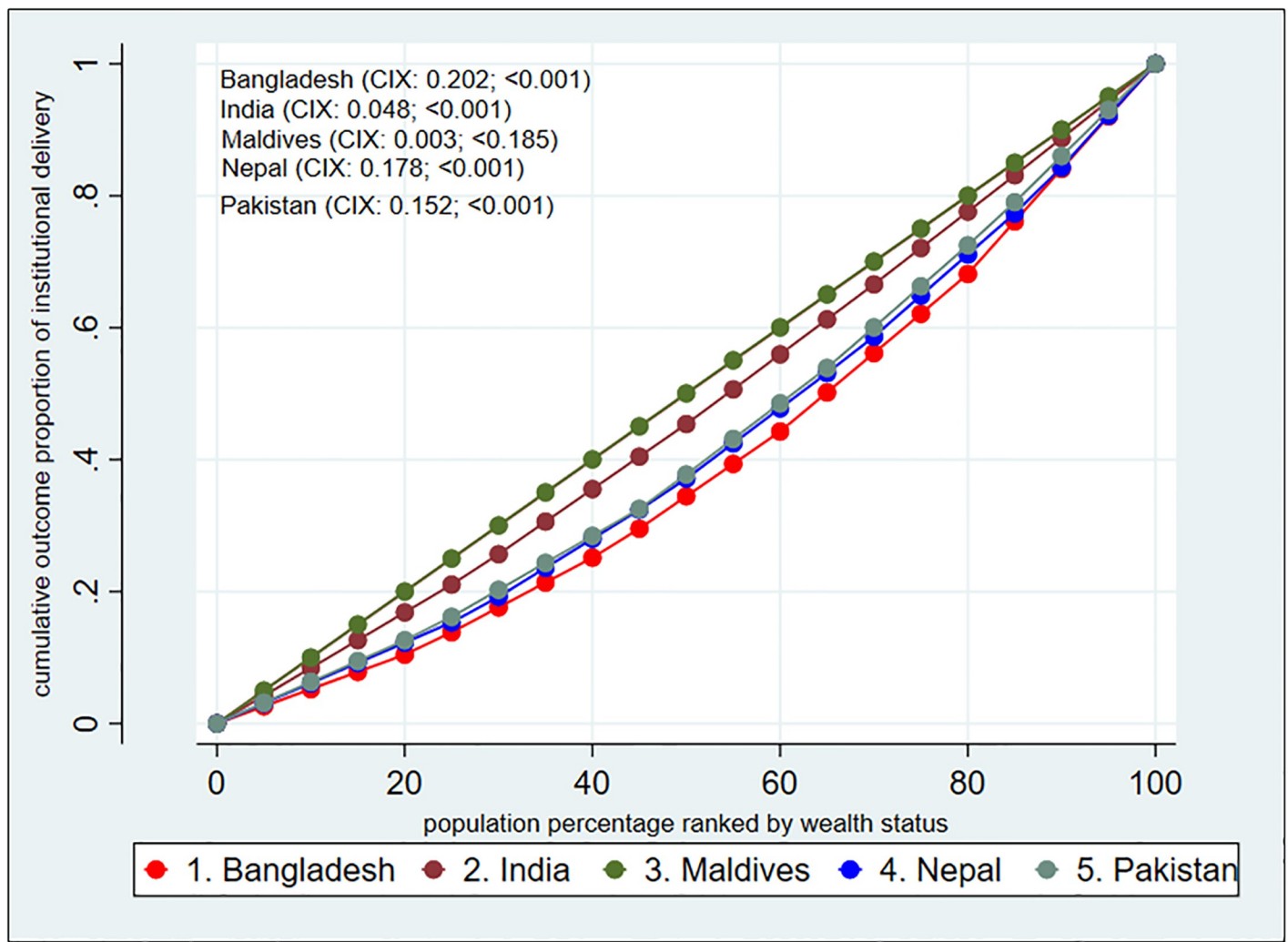

**Fig 3. Concentration curve for inequality estimation of institutional delivery behavior.**

Elasticity with a positive or negative sign indicates a positive change in the factor with a rising or falling trend in the facility's output. The values of CIX that were calculated in this analysis to determine the level of socioeconomic inequity have values between -1 and +1. However, if the facilities are equally distributed among the socio-economic groups, the CIX value becomes 0. In this study, the value of the CIX for maternal health care utilization was found to be (ANC: 0.2346; p<0.001 and institutional delivery: 0.3125; p<0.001) among women with a higher socioeconomic status, suggesting socioeconomic inequality in utilizing maternal health care in favor of the more affluent. The column 'CIX' displays the distribution of the determinants in terms of wealth quintiles. Whether the factors were more prominent in the affluent or poor group is shown by the positive or negative direction of the CIX. The percentage contribution shows how much each model variable adds to overall socioeconomic differences in utilizing maternal health care. A component that is anticipated to widen socioeconomic gaps connected to maternal health care usage has a positive percentage contribution. On the other hand, a factor with a negative percentage contribution suggests a drop in socioeconomic disparities in the use of maternal health care facilities.

**Table 3. Decomposition analysis of ANC seeking behaviour: Pooled analysis.**

| | Elasticity | CIX | Contribution to overall CIX = 0.2346 (p<0.001) | |
| --- | --- | --- | --- | --- |
| | | | Absolute contribution | Percentage contribution |
| **Characteristics** | | | | |
| **Name of the Country** | | | | |
| Bangladesh | -0.0123 | 0.0609 | -0.0007 | -0.6982 |
| India | 0.1548 | -0.0050 | -0.0008 | -0.7149 |
| Maldives | 0.0286 | 0.0459 | 0.0013 | 1.2232 |
| Nepal | 0.0111 | 0.0262 | 0.0003 | 0.2708 |
| Pakistan (RC) | | | | |
| *Sub-total* | | | *0.0001* | *0.0809* |
| **Place of Residence** | | | | |
| Urban | 0.0521 | 0.4316 | 0.0225 | 20.9906 |
| Rural (RC) | | | | |
| *Sub-total* | | | | *20.9906* |
| **Maternal Age** | | | | |
| 15–24 | 0.0076 | -0.0721 | -0.0005 | -0.5095 |
| 25–34 | 0.0094 | 0.0449 | 0.0004 | 0.3954 |
| 35–49 (RC) | | | | |
| *Sub-total* | | | *-0.0001* | *-0.1141* |
| **Body Mass Index** | | | | |
| <18.50 (Underweight) | -0.0145 | -0.2026 | 0.0029 | 2.7322 |
| 18.50–24.90 (Normal) (RC) | | | | |
| 25.00–29.99 (Overweight) | 0.0268 | 0.2586 | 0.0069 | 6.4646 |
| <30 (Obesity) | 0.0101 | 0.3761 | 0.0038 | 3.5425 |
| *Sub-total* | | | *0.0136* | *12.7393* |
| **Women's Highest Education Level** | | | | |
| No education (RC) | | | | |
| Primary | 0.0423 | -0.2181 | -0.0092 | -8.6001 |
| Secondary | 0.2739 | 0.0706 | 0.0193 | 18.0315 |
| Higher | 0.1140 | 0.5099 | 0.0581 | 54.2215 |
| *Sub-total* | | | *0.0682* | *63.6529* |
| **Current Working Status** | | | | |
| Not working (RC) | | | | |
| Working | 0.0303 | -0.0988 | -0.0030 | -2.7924 |
| **Husband's Education Level** | | | | |
| No education (RC) | | | | |
| Primary | 0.0393 | -0.2143 | -0.0084 | -7.8588 |
| Secondary | 0.1321 | 0.0675 | 0.0089 | 8.3139 |
| Higher | 0.0461 | 0.4373 | 0.0202 | 18.7988 |
| *Sub-total* | | | *0.0207* | *19.2539* |
| **Husband's Occupation** | | | | |
| Agricultural (RC) | | | | |
| Non-Agricultural | 0.0653 | 0.0053 | 0.0003 | 0.3200 |
| **Wealth Status** | | | | |
| Poorest (RC) | | | | |
| Poorer | 0.0485 | -0.2980 | -0.0144 | -13.4715 |
| Middle | 0.0898 | 0.1144 | 0.0103 | 9.5828 |

(*Continued*)

**Table 3.** (Continued)

| | Elasticity | CIX | Contribution to overall CIX = 0.2346 (p<0.001) | |
| --- | --- | --- | --- | --- |
| | | | Absolute contribution | Percentage contribution |
| Richer | 0.0912 | 0.4961 | 0.0453 | 42.2116 |
| Richest | 0.1204 | 0.8407 | 0.1012 | 94.3946 |
| *Sub-total* | | | 0.1424 | *132.7175* |
| *Explained CIX* | | | 0.2647 | |
| *Residual CIX* | | | -0.0301 | |

## Discussion

Despite massive efforts and initiatives by local, national, and international organizations for the provision of maternal health care services, still a substantial volume of people in several south Asian countries remain unutilized. Nonetheless, in a number of south Asian nations, such as Bangladesh, Nepal, Malaysia, and Sri Lanka, the number of women receiving care-related services has increased dramatically in recent years, demonstrating impressive development [20]. To meet SDG's targets of reducing maternal mortality to 70 per 100,000 live births by 2030, various south Asian nations, notably Bangladesh, India, and Pakistan, must improve their maternal health care facilities' quality and accessibility [5,20]. These findings may assist the government and other interested parties in planning, designing, and executing the essential measures, as well as removing obstacles to enhancing health care utilization and, thus, reducing maternal mortality in those nations. This study examines a number of hypothesized parameters associated with the usage of maternity care services. Women from urban areas in all four countries are more likely to use maternal care services, a finding borne up by a large number of prior research [11,25,32–35]. A plausible explanation might be that urban women may have certain socioeconomic advantages over rural women. Urban women are more educated, more knowledgeable, and have easier access to both public and private health-care facilities, whereas rural women may not have the same opportunities [4,11,15,36]. In addition, urban areas are better equipped with media facilities, which can play a vital role in improving maternal health care service utilization. In a study of India [11] and other settings, reached same conclusions [16,33]. As a result, ANC visits would become more frequent enhancing institutional delivery. This inequality in access to healthcare must be reduced, however, and this can only be accomplished through systematic measures.

Women's and husbands' educational attainment were discovered as a significant predictor of the use of both types of services for maternal health care in all countries. However, rather than the education of the husband, women's education has a greater influence on their decision to utilize maternal health care services. According to previous research undertaken in lower- and middle-income countries (LMICs), women with greater educational attainments are considerably more likely to utilize maternal care services than women without any formal schooling [11,15,16,36,37]. Additionally, it is clearly illustrated that women's academic achievement has a major effect on their utilization of maternal health care. It implies that women who are well-informed are more likely to be educated about health-related information and aware of the negative impacts of postponing maternity care. While educated husbands are more likely to connect with their wives more effectively and be more receptive to discussing maternal health services, they also enhance the value of maternity care services [15]. As a result, the research proposes to facilitate educational facilities for all in order to improve access to and usage of maternal health care facilities.

**Table 4. Decomposition analysis of institutional delivery: Pooled analysis.**

| Characteristics | Elasticity | CIX | Contribution to overall CIX = 0.3125 (p<0.001) | |
| --- | --- | --- | --- | --- |
| | | | Absolute contribution | Percentage contribution |
| **Name of the Country** | | | | |
| Bangladesh | -0.0212 | 0.0609 | -0.0013 | -2.4323 |
| India | 0.4948 | -0.0050 | -0.0025 | -4.6130 |
| Maldives | 0.0181 | 0.0459 | 0.0008 | 1.5604 |
| Nepal | -0.0070 | 0.0262 | -0.0002 | -0.3431 |
| Sub-total | | | -0.0032 | -5.828 |
| Pakistan (RC) | | | | |
| **Place of Residence** | | | | |
| Urban | 0.0230 | 0.4316 | 0.0099 | 18.6469 |
| Rural (RC) | | | | |
| **Maternal Age** | | | | |
| 15–24 | 0.0541 | -0.0721 | -0.0039 | -7.3481 |
| 25–34 | 0.0294 | 0.0449 | 0.0013 | 2.4865 |
| 35–49 (RC) | | | | |
| Sub-total | | | -0.0026 | -4.8616 |
| **Body Mass Index** | | | | |
| <18.50 (Underweight) | -0.0023 | -0.2026 | 0.0005 | 0.8667 |
| 18.50–24.90 (Normal) (RC) | | | | |
| 25.00–29.99 (Overweight) | 0.0150 | 0.2586 | 0.0039 | 7.3182 |
| <30 (Obesity) | 0.0103 | 0.3761 | 0.0039 | 7.2635 |
| Sub-total | | | 0.0083 | 15.4484 |
| **Women Highest Education** | | | | |
| No education (RC) | | | | |
| Primary | 0.0205 | -0.2181 | -0.0045 | -8.4258 |
| Secondary | 0.1827 | 0.0706 | 0.0129 | 24.2854 |
| Higher | 0.1039 | 0.5099 | 0.0530 | 99.7738 |
| Sub-total | | | 0.0614 | 115.6334 |
| **Current Working Status** | | | | |
| Not working (RC) | | | | |
| Working | -0.0009 | -0.0988 | 0.0001 | 0.1749 |
| **Husband's Education Level** | | | | |
| No education (RC) | | | | |
| Primary | 0.0144 | -0.2143 | -0.0031 | -5.8251 |
| Secondary | 0.0641 | 0.0675 | 0.0043 | 8.1412 |
| Higher | 0.0438 | 0.4373 | 0.0191 | 36.0111 |
| Sub-total | | | 0.0203 | 38.3272 |
| **Husband's Occupation** | | | | |
| Agricultural (RC) | | | | |
| Non-Agricultural | 0.0268 | 0.0053 | 0.0001 | 0.2649 |
| **Wealth Status** | | | | |
| Poorest (RC) | | | | |
| Poorer | 0.0498 | -0.2980 | -0.0149 | -27.9572 |
| Middle | 0.0715 | 0.1144 | 0.0082 | 15.4057 |
| Richer | 0.0946 | 0.4961 | 0.0469 | 88.3604 |

(*Continued*)

**Table 4.** (Continued)

| | Elasticity | CIX | Contribution to overall CIX = 0.3125 (p<0.001) | |
| --- | --- | --- | --- | --- |
| | | | Absolute contribution | Percentage contribution |
| Richest | 0.1176 | 0.8407 | 0.0988 | 186.0299 |
| Sub-total | | | 0.139 | 261.8388 |
| Explained CIX | | | 0.2333 | |
| Residual CIX | | | 0.0792 | |

Aligning with earlier studies, this study showed that usage of both two type of maternity services in south Asian countries is strongly connected with household wealth status [11,24,25,33,36,37]. It would be because economically affluent women would choose a better hospital facility and would have easier access to health care information. In Bangladesh, India, and Nepal, we noticed that women who ended up working in income-generating pursuits are much less likely to use institutional delivery services. Even though working women might have the financial means to pay for healthcare, there is little correlation between the two, which is not in favour for institutional delivery but favour for ANC. This is in line with earlier researches, which also contends that autonomy and decision-making capacity play critical roles in maternal health care-seeking behavior [38–41]. Though negative relations between institutional delivery and work involvement stand against empirical evidence, it can be because of inadequate institutional facilities provided by respective countries or working pressure, which is unclear to researchers, therefore, further study is recommended in this regard to point it out.

In India and Pakistan, women between the ages of 15 and 24 were more likely to use maternity care facilities, whereas maternal age is not important in Bangladesh and Nepal. Early pregnancy increases the risk of viral and parasitic infections and pregnancy-related problems, and the absence of hemoglobinopathies that cause anemia may push women to seek maternity care facilities in India and Pakistan [4,22]. However, this is not consistent with the previous study would be because of contextualizing and demographic ethnicity reasons [11].

Additionally, this study discovered a substantial correlation between BMI and the utilization of maternity services across all four nations. Compared to women with normal BMI, individuals with high BMI consistently show a higher risk of accessing maternal healthcare services. This result is consistent with a few earlier research [42–44], and one of the main reasons behind this scenario is that people with high BMI are more likely to have other complexities or the probability of encountering complexities during or before delivery is high [45,46]. They must therefore use maternal healthcare services more frequently than their counterparts. However, there are notable regional differences in maternity care due to socioeconomic level, education, location, and access to health care resources.

This study's strength was, in fact, its utilization of the most recently published national demographic data. The surveys were conducted at the level of the population of each county. Due to the breadth of the sample, the results may be extrapolated to a wide range of situations. Along with appropriate statistical methods and decomposition of the inequality measure and Lorenz estimates were performed using standard approaches. Due to the cross-sectional structure of the survey, no causal relationships can be inferred, as exposure/predictors and outcomes are evaluated concurrently.

## Conclusion and policy implications

This study compares maternal health care services across south Asian nations and offers some original findings as well as those that are consistent with those of other published studies. The results of this study thus suggested broad views for outlining the overall strategy for increasing the utilization of maternity care services.

Relentless importance should be given to ensure timely and proper ANC utilization with at least four visits because there are big disparities between the target and reality. Increase the proper postpartum care centers where women can get pre-delivery and after-delivery services to reduce the complexities that arise during pregnancy. Take strategic measures to effectively reduce socioeconomic inequalities among reproductive-aged women in using pregnancy care, because this study points out that income, education, partners' education, location etc., have significant effects on using maternal health care services. Expanding funds for education because it plays a vital role in awareness building and also parallel to media exposure, all of which have positive effects on utilizing maternal care services. Ensuring proper freedom of media and women empowerment to positively influence women to utilize maternal health care services.

However, these plans and tactics can only be successful if there is sufficient coordination and political commitment across all parties. Intensified community engagement, fair distribution of finances, appropriate technology, and distribution equity all serve as intensifiers in the process of employing maternity care services. Fundamentally creating an enabling environment for all reproductive women is the real challenge; if the government can achieve a perfect environment or at least can crack the code of enabling, only then we can ensure proper maternal health.

## Supporting information

**S1 Table. Factors associated with ANC and institutional delivery: Bangladesh.**
(DOCX)

**S2 Table. Factors associated with ANC and institutional delivery: India.**
(DOCX)

**S3 Table. Factors associated with ANC: Institutional delivery: Nepal.**
(DOCX)

**S4 Table. Factors associated with ANC: Institutional delivery: Pakistan.**
(DOCX)

## Acknowledgments

We thank the MEASURE DHS Data Archive, ICF International, for providing access to the South and Southeast Asian Demographic and Health Surveys data.

## Author Contributions

**Conceptualization:** Md. Ashfikur Rahman, Md. Sazedur Rahman.

**Data curation:** Md. Ashfikur Rahman, Md. Amirul Islam.

**Formal analysis:** Md. Ashfikur Rahman.

**Methodology:** Md. Ashfikur Rahman, Mortuja Mahamud Tohan.

**Supervision:** S. M. Muhibullah, Md. Hasan Howlader.

**Visualization:** Md. Ashfikur Rahman.

**Writing – original draft:** Md. Ashfikur Rahman, Md. Amirul Islam, Mortuja Mahamud Tohan.

**Writing – review & editing:** Mortuja Mahamud Tohan, S. M. Muhibullah, Md. Sazedur Rahman, Md. Hasan Howlader.

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
