## [Decision Letter · Decision Letter 0]

11 Oct 2023

PONE-D-23-26546Socioeconomic Inequalities in Utilizing Maternal Health Care in Five South Asian Countries: A Decomposition AnalysisPLOS ONE

Dear Dr. Rahman,

Thank you for submitting your manuscript to PLOS ONE. After careful consideration, we feel that it has merit but does not fully meet PLOS ONE’s publication criteria as it currently stands. Therefore, we invite you to submit a revised version of the manuscript that addresses the points raised during the review process.

We look forward to receiving your revised manuscript.

Kind regards,

Satyajit Kundu

Academic Editor

PLOS ONE

Journal Requirements:

Additional Editor Comments:

The authors are requested to address all the comments from reviewers carefuly.

Reviewers' comments:

Reviewer's Responses to Questions

**Comments to the Author**

1. Is the manuscript technically sound, and do the data support the conclusions?

Reviewer #1: Yes

Reviewer #2: Yes

2. Has the statistical analysis been performed appropriately and rigorously? 

Reviewer #1: Yes

Reviewer #2: Yes

3. Have the authors made all data underlying the findings in their manuscript fully available?

Reviewer #1: Yes

Reviewer #2: Yes

4. Is the manuscript presented in an intelligible fashion and written in standard English?

Reviewer #1: Yes

Reviewer #2: No

5. Review Comments to the Author

Reviewer #1: Thanks to the authors for addressing an important topic and I am glad to review the paper. The paper is sound interesting and important, although lots of work already done on the same topic using secondary DHS dataset. Please justify your argument why you research is important.

I have some specific thoughts that needs to be addressed for publication and reaching to wider community.

• Rationality of the paper is very week, please focus on the research gaps that is missing in the introduction section.

• What is robustness of the analysis, why authors choose this statistical analysis?

• Some grammatical errors found please check the whole manuscript.

• What is the policy implications and theoretical contribution of this study?

• Why did you choose these five countries only while there are other South Asian countries have?

• How you choose the independent variables for final model that is missing?

• What is power of concentration index and elasticity analysis?

• I would suggest you to edit the conclusion based on the findings.

• Provide a calculation of explained and residual CIX in your decomposition table.

Reviewer #2: The objectives of this study were to assess socioeconomic disparities in the utilization of health care services by

mothers and to determine the factors influencing this utilization among women aged 15 to 49 in five South Asian countries. This study has important policy implications. The overall quality of this manuscript is good. However, the quality will improve through linguistic and grammatical improvements. Manuscript's similarity index should be checked to ensure uniqueness. The result section should be concise, reduce background characteristics. Would you please enrich the strengths and limitations section?

6. PLOS authors have the option to publish the peer review history of their article (what does this mean?). If published, this will include your full peer review and any attached files.

Reviewer #1: No

Reviewer #2: No

---

## [Author Response · Author response to Decision Letter 0]

13 Oct 2023

All comments have been addressed and provided a rebuttal.

---

## [Decision Letter · Decision Letter 1]

26 Oct 2023

PONE-D-23-26546R1Socioeconomic Inequalities in Utilizing Maternal Health Care in Five South Asian Countries: A Decomposition AnalysisPLOS ONE

Dear Dr. Rahman,

Thank you for submitting your manuscript to PLOS ONE. After careful consideration, we feel that it has merit but does not fully meet PLOS ONE’s publication criteria as it currently stands. Therefore, we invite you to submit a revised version of the manuscript that addresses the points raised during the review process.

We look forward to receiving your revised manuscript.

Kind regards,

Satyajit Kundu

Academic Editor

PLOS ONE

Journal Requirements:

Additional Editor Comments:

- The paper addressed all the comments and now it can be acceptable after doing a minor change. Provide the values for sub-total of each variable in the decomposition tables of both outcome variables. Then give the summarized explained, and residual CIX contribution just beneath the “absolute contribution” column in Table 3 and Table 4.

Reviewers' comments:

Reviewer's Responses to Questions

**Comments to the Author**

1. If the authors have adequately addressed your comments raised in a previous round of review and you feel that this manuscript is now acceptable for publication, you may indicate that here to bypass the “Comments to the Author” section, enter your conflict of interest statement in the “Confidential to Editor” section, and submit your "Accept" recommendation.

Reviewer #1: All comments have been addressed

Reviewer #2: All comments have been addressed

2. Is the manuscript technically sound, and do the data support the conclusions?

Reviewer #1: Yes

Reviewer #2: Yes

3. Has the statistical analysis been performed appropriately and rigorously? 

Reviewer #1: Yes

Reviewer #2: I Don't Know

4. Have the authors made all data underlying the findings in their manuscript fully available?

Reviewer #1: Yes

Reviewer #2: Yes

5. Is the manuscript presented in an intelligible fashion and written in standard English?

Reviewer #1: Yes

Reviewer #2: Yes

6. Review Comments to the Author

Reviewer #1: I am satisfied to accept the paper. All comments are clearly addressed. So, i believe the paper is in a acceptable format.

Reviewer #2: The manuscript is technically sound and the data supports the conclusions. I have no further comments.

7. PLOS authors have the option to publish the peer review history of their article (what does this mean?). If published, this will include your full peer review and any attached files.

Reviewer #1: No

Reviewer #2: No

---

## [Author Response · Author response to Decision Letter 1]

14 Dec 2023

All comments have been addressed.

---

## [Editor Report · Decision Letter 2]

18 Dec 2023

Socioeconomic Inequalities in Utilizing Maternal Health Care in Five South Asian Countries: A Decomposition Analysis

PONE-D-23-26546R2

Dear Dr. Ashfikur Rahman,

We’re pleased to inform you that your manuscript has been judged scientifically suitable for publication and will be formally accepted for publication once it meets all outstanding technical requirements.

Kind regards,

Satyajit Kundu, MSc

Academic Editor

PLOS ONE

Additional Editor Comments (optional):

After carefully review the revision of your paper, I am pleased to let you know that the paper is scientifically proof now.

---

## [Editor Report · Acceptance letter]

1 Feb 2024

PONE-D-23-26546R2 

PLOS ONE

Dear Dr. Rahman, 

I'm pleased to inform you that your manuscript has been deemed suitable for publication in PLOS ONE. Congratulations! Your manuscript is now being handed over to our production team.

Kind regards, 

on behalf of

Dr. Satyajit Kundu 

Academic Editor

PLOS ONE